# Effects of Sequential Enzymolysis and Glycosylation on the Structural Properties and Antioxidant Activity of Soybean Protein Isolate

**DOI:** 10.3390/antiox12020430

**Published:** 2023-02-09

**Authors:** Qing Zhang, Lin Li, Lan Chen, Shuxiang Liu, Qiang Cui, Wen Qin

**Affiliations:** Key Laboratory of Agricultural Product Processing and Nutrition Health of Ministry of Agriculture and Rural Affairs (Co-Construction by Ministry and Province), College of Food Science, Sichuan Agricultural University, No. 46, Xinkang Road, Ya’an 625014, China

**Keywords:** soybean protein isolate, enzymolysis, glycosylation, structural properties, antioxidant activity

## Abstract

The effects of limited hydrolysis following glycosylation with dextran on the structural properties and antioxidant activity of the soybean protein isolate (SPI) were investigated. Three SPI hydrolysate (SPIH) fractions, F30 (>30 kDa), F30-10 (10–30 kDa), and F10 (<10 kDa), were confirmed using gel permeation chromatography. The results demonstrated that the glycosylation of F30 was faster than that of F30-10 or F10. The enzymolysis caused the unfolding of the SPI to expose the internal hydrophobic cores, which was further promoted by the grafting of dextran, making the obtained conjugates have a loose spatial structure, strong molecular flexibility, and enhanced thermal stability. The grafting of dextran significantly enhanced the DPPH radical or •OH scavenging activity and the ferrous reducing power of the SPI or SPIH fractions with different change profiles due to their different molecular structures. The limited enzymolysis following glycosylation was proven to be a promising way to obtain SPI-based food ingredients with enhanced functionalities.

## 1. Introduction

Plant proteins have gained increasing attention for food consumption because of their advantages in sustainability and health, as well as ethical reasons [1]. However, the structural and functional properties of plant proteins during processing are easily negatively affected by various factors, such as pH, high temperature, and salt, limiting their industrial applications. Therefore, the modification of functionalities has become a mainstream for developing plant protein-based foods. Physical, chemical, and enzymatic methods are developed to modify the structure of proteins and, thus, improve their techno-functionalities. The proteins modified by a single method can acquire improved functional properties; however, these effects always seem to be insufficient. The physical methods cause relatively less destruction to the structure or nutritional value of the native proteins, but the improvements of the functional properties are minimal [2]. Although chemical methods are commonly used to obtain modified proteins, many drawbacks, such as the use of chemicals, site-nonspecific modification, and the possible occurrence of undesirable byproducts, are found [3]. Enzymolysis is a site-specific and mild reaction; however, the control of the reaction process and the flavor of end products are also challenging for the commercial application of this method [4]. Therefore, the combination of two or more methods has been recently applied to modify the structure and functional properties of proteins [5].

Glycosylation, also known as the first stage of the Maillard reaction, combines the macromolecular characteristics of proteins and hydrophilicity of reducing saccharides via covalent bonds, conferring to the product’s excellent solubility, emulsifying properties, gelling properties, thermal stability, and viscoelasticity [6]. Glycosylation combined with enzymolysis provides an enlightening strategy for the further development of protein resources to satisfy the demand of special food ingredients. Jiang et al. [7] reported that the glycosylation of whey protein isolate (WPI) with galactose following enzymolysis exhibited a higher hydrolysis degree and antioxidant activity than the native or heated WPI, suggesting that synergistic modification is feasible. A previous study regarding first glycosylation, and then enzymolysis of soybean protein isolate (SPI), showed that the SPI-oligochitosan conjugate hydrolysate exhibited improved emulsifying activity and antioxidant activities due to an appropriate decomposition of conjugate molecules [8].

First the enzymolysis and then the glycosylation of proteins was also previously reported. WPIs first hydrolyzed and then conjugated with lactose or lactulose demonstrated that hydrolyzed WPI-sugar conjugates possessed higher DPPH radical scavenging activity than WPI-sugar conjugates [9]. Xiao et al. [10] reported that the grafting of glucose onto a rice dreg protein improved the antioxidant activity of protein hydrolysates and the trypsin-hydrolyzed protein-glucose conjugate exhibited the highest DPPH radical scavenging activity. These studies indicate that an appropriate enzymolysis of protein before glycosylation also provides a better modification of functionalities than solely glycosylation.

It can be concluded that the combination of enzymolysis and glycosylation could confer to proteins improved functionalities based on the changes in the molecular structure. Although SPI is the most consumed vegetable protein ingredient in the current food industry, its functionalities are generally weakened due to the intrinsic drawbacks of the existing manufacturing process. As a neutral polysaccharide, dextran exhibits many advantages, such as the low cost, extensive sources, wild range of molecular weight (MW), and easy operation to perform protein modification via glycosylation. In addition, most previous studies selected a wet-heating method to perform glycosylation. Therefore, we performed an initial enzymolysis and then glycosylation of SPI with dextran using a dry-heating method to investigate the changes in the molecular structure and antioxidant activities in this study, aiming to develop innovative protein-based food ingredients by combined enzymolysis and glycosylation.

## 2. Materials and Methods

### 2.1. Materials and Chemicals

The SPI (95.86% crude protein content, measured by the Kjeldahl method) was purchased from Solarbio Science & Technology Co., Ltd. (Beijing, China). The neutrase (BR, 100 U/mg) and dextran (MW 4 kDa) were purchased from Yuanye Biotechnology Co., Ltd. (Shanghai, China). The sodium dodecyl sulfate-polyacrylamide gel electrophoresis (SDS-PAGE) gel preparation kit was bought from Boster Biological Technology Co., Ltd. (Wuhan, China). A molecular marker (11–245 kDa) for protein composition analysis was bought from Tiangen Biotech Co., Ltd. (Beijing, China). All other chemicals were of an analytical grade and obtained from Chengdu Chron Chemical Co., Ltd. (Chengdu, China).

### 2.2. Preparation of SPI Hydrolysates (SPIH)

The enzymolysis of SPI was conducted according to our previous method [11]. Briefly, SPI powder was initially dispersed in ultrapure water (5%, *w*/*v*), magnetically stirred at an ambient temperature for 30 min, and then kept overnight at 4 °C. After adjusting the pH to 7.0 using 0.1 M NaOH or HCl, the SPI dispersion was preheated at 55 °C in a water bath for 30 min. Thereafter, the SPI dispersion was partially hydrolyzed by adding 0.5% (*w*/*w*) neutrase, followed by mild stirring at 120 rpm and 55 °C for 20 min. The enzymatic solution was then heated at 90 °C for 20 min. After cooling to room temperature in an ice water, the solution was centrifuged (RJ-TGL-2000R, Ruijiang Analysis Instrument Co., Ltd., Wuxi, China) at 9030× *g* and 4 °C for 10 min, and the supernatant was filtered using 0.45 μm water membrane filters to collect SPIH.

### 2.3. Ultrafiltration of SPIH

The ultrafiltration of SPIH was conducted using Amicon Ultra-15 Centrifugal filters (Millipore, Merck KgaA, Darmstadt, Germany) according to our previous study [11]. The SPIH was first fractionated by a 30 kDa MW cutoff membrane at 1440× *g* and 4 °C for 20 min to obtain a retentate (F30) and permeate 1. The permeate 1 was further filtered using a 10 kDa MW cutoff membrane to obtain another retentate (F30-10) and a second permeate (F10). Therefore, the MW of F30, F30-10, and F10 were >30, 10–30, and <10 kDa, respectively. These three fractions were individually freeze-dried using a vacuum freeze dryer (LGJ-18S, Ningbo Xinyi Ultrasonic Equipment Co., Ltd., Ningbo, China), and stored at −20 °C until analysis. The yields of F30, F30-10, and F10 were calculated as 15.315%, 3.941%, and 0.843%, respectively.

### 2.4. Determination of MW Distribution and Protein Composition of SPIH Fractions

The MW distribution of F30, F30-10, and F10 was measured using a gel permeation chromatography (GPC, Waters 1515) with a Waters GPC column and a 2414 differential refractive index detector. Briefly, the aqueous solution of F30, F30-10, or F10 (5 mg/mL) was eluted in a gradient mode with a sodium nitrate solution (0.1 mol/L) as a mobile phase and a speed of 1 mL/min. The temperature of the column and detector was 40 °C.

SDS-PAGE was performed using vertical gel electrophoresis equipment (Mini-Protein, Bio-Rad Laboratories, Inc., Hercules, CA, USA) with a 12% (*w*/*v*) separating gel and a 5% (*w*/*v*) stacking gel [11]. SPI, SPIH, F30, F30-10, and F10 were dissolved in a 2-fold buffer (pH 6.8) with a concentration of 3 mg/mL, and then boiled for 3 min. The buffer was prepared by mixing 0.1 mol/L Tris-HCl, 4% SDS, 10% glycerol, 0.005% bromophenol blue, and 10% β-mercaptoethanol. The sample solution was centrifuged at 9570× *g* for 10 min. Aliquots (10 μL) of the supernatant were added to the wells and electrophoresis was performed at 120 V. After separating, the gels were stained with 0.25% (*w*/*v*) Coomassie Brilliant Blue R-250, and then de-stained using 25% methanol and 10% acetic acid for scanning, using an image scanner (Smart Gel 130, Saizhi Scientific Co., Ltd., Beijing, China).

### 2.5. Preparation of SPIH-Dextran Conjugates

F30, F30-10, and F10 grafting with dextran were performed using a dry-heating method [11]. F30, F30-10, or F10 and dextran (1:1, *w*/*w*) were suspended in ultrapure water at a protein concentration of 2% (*w*/*v*), while stirring at room temperature for 10 min. After the pH was adjusted to 7.0 with 0.1 M NaOH or HCl, the dispersion was then lyophilized. The resulting powder was incubated at 60 °C and 79% relative humidity for 24 h. After the incubation, the powder was cooled to terminate the reaction and then stored at −20 °C prior to analysis. The mixtures of F30, F30-10, or F10 and dextran without thermal treatment were prepared as controls.

### 2.6. Confirmation of F30, F30-10, or F10-Dextran Conjugates

The formation of F30, F30-10, or F10-dextran conjugates was confirmed by measuring the amount of Amadori compounds and browning degree and analyzing the changes in the protein composition using the electrophoretic analysis mentioned above.

The amounts of Amadori compounds and browning degree were determined according to a method proposed by Wang and Ismail [12]. The samples were suspended in 0.01 mol/L pH 7.2 phosphate buffer solution (PBS) at a protein concentration of 2 mg/mL and then centrifuged at 9570× *g* for 10 min. The absorbance values of the supernatants at 304 nm and 420 nm (UV-visible spectrophotometer, UV-6100, Metash Instruments Co., Ltd., Shanghai, China) were defined as indications of Amadori compounds and the browning degree, respectively.

The method used to complete the electrophoretic analysis of conjugates was the same as that mentioned above, except that the separation of F10 and F10-dextran conjugate was performed using a 15% separating gel.

### 2.7. Structural Analysis of F30, F30-10, or F10-Dextran Conjugates

#### 2.7.1. Ultraviolet (UV)-Visible Spectroscopy

According to a previous method [13], SPI, SPIH, F30, F30-10, and F10, and F30, F30-10, or F10-dextran conjugates were individually dissolved in 0.01 mol/L PBS (pH 7.2) to prepare a sample solution with a protein concentration of 2 mg/mL. After vortical mixing for 5 min, the sample solution was maintained at 4 °C overnight. Thereafter, the sample solution was centrifuged at 9570× *g* for 10 min and the supernatant was then subjected to ultraviolet spectrum scanning with a wavelength range of 270–400 nm and a wavelength interval of 2 nm at an ambient temperature.

#### 2.7.2. Intrinsic Fluorescence Emission Spectroscopy

The intrinsic emission fluorescence spectra were measured using a Lumina fluorescence spectrometer (Thermo Fisher Scientific^®^, Waltham, MA, USA). The sample preparation was the same as that described in Section 2.7.1. The supernatant was excited at 290 nm and the emission spectra were recorded from 270 to 500 nm, with a wavelength interval of 2 nm at the ambient temperature.

#### 2.7.3. Fourier Transform Infrared (FTIR) Spectroscopy

The FTIR spectra were recorded from salt disks consisting of 2 mg of samples and 200 mg of potassium bromide (KBr) using an FTIR spectrometer (NicoletlS10, Thermo Fisher Scientific^®^, Waltham, MA, USA). The background noise was corrected by a pure KBr disk. The scanning was carried out within a wavenumber range of 4000–400 cm^−1^ at a resolution of 4 cm^−1^, and 16 scans were averaged for each sample.

#### 2.7.4. Differential Scanning Calorimetry (DSC)

The thermal denaturation properties of the samples were studied using a TA instrument controlled by a TA 2000 system (Q200M, TA Instruments, New Castle, DE, USA). The freeze-dried sample (6 mg) was hermetically sealed in aluminum pans and then equilibrated at an ambient temperature for 2 h. An empty aluminum pan was used as a reference. A temperature ramp from 20 to 250 °C was scanned at a rate of 5 °C/min under an inert atmosphere (50 mL/min of N_2_). The peak temperature (*T*_p_) and total calorimetric apparent enthalpy change (Δ*H*) were analyzed using a heat absorption curve.

### 2.8. Measurements of Antioxidant Activity

The antioxidant activities of SPI, SPIH, F30, F30-10, and F10, and F30, F30-10, or F10-dextran conjugates were evaluated by analyzing the DPPH radical scavenging activity, hydroxyl free radical (•OH) scavenging activity, and ferrous reducing power.

#### 2.8.1. Determination of DPPH Radical Scavenging Activity

The sample preparation procedure was the same as that described in Section 2.7.1. One milliliter supernatant was mixed with 2 mL 0.1 mmol/L DPPH radical-ethanolic solution. After staying at 37 °C in a dark environment for 30 min, the absorbance of the resulting solution was measured at 517 nm using a UV-visible spectrophotometer. Under the same treatment manner, the blank group used distilled water instead of the sample solution and the control group used anhydrous ethanol rather than the DPPH solution. Ascorbic acid at a concentration of 0.01 mg/mL was used as a positive control. The DPPH radical scavenging activity was calculated by Equation (1):(1)$$\mathrm{DPPH}\;\mathrm{radical}\;\mathrm{scavenging}\;\mathrm{activity}\;(\%)=\left[1-\frac{A_{s}-A_{c}}{A_{b}}\right]\times 100{\%}_{}$$
where *A_s_*, *A_b_*, and *A_c_* represent the absorbance of the sample, blank, and control, respectively.

#### 2.8.2. Determination of •OH Scavenging Activity

After preparation according to the procedure described in Section 2.7.1., 1 mL of supernatant was added with 1 mL of the ferrous sulfate solution (6 mmol/L) and 1 mL of the salicylic acid ethanolic solution (6 mmol/L). After thoroughly mixing, 1 mL hydrogen peroxide solution was added to initiate the reaction. After incubating at 37 °C in a water bath for 30 min, the absorbance of the resulting solution was measured at 510 nm. Under the same treatment manner, the blank group used distilled water instead of the sample solution and the control group used distilled water rather than the hydrogen peroxide solution. Ascorbic acid at a concentration of 0.01 mg/mL was used as a positive control. The •OH scavenging activity was calculated by Equation (2):(2)$$•\mathrm{OH}\;\mathrm{scavenging}\;\mathrm{activity}\;(\%)=\left[1-\frac{A_{s}-A_{c}}{A_{b}}\right]\times 100{\%}_{}$$
where *A_s_*, *A_b_*, and *A_c_* represent the absorbance of the sample, blank, and control, respectively.

#### 2.8.3. Determination of Ferrous Reducing Power

After preparation according to the procedure described in Section 2.7.1., 1 mL of supernatant was successively added with 1 mL 0.2 mol/L PBS (pH 6.6) and 1 mL potassium ferricyanide solution (1%, *w*/*v*). After maintaining at 50 °C in a water bath for 30 min, an aliquot (1.0 mL) of 10% trichloroacetic acid (TCA) was added to stop the reaction. Then, 1 mL of the reactant solution was mixed with 1 mL of the distilled water and 0.2 mL of ferric trichloride (0.1%, *w*/*v*), and the absorbance of the resulting solution was recorded at 700 nm. The deionized water was used as a blank instead of a sample solution. The stronger absorbance of the mixture indicated an increasing ferrous reducing power. The ascorbic acid at a concentration of 0.01 mg/mL was used as a positive control.

### 2.9. Statistical Analysis

All experiments were conducted in triplicate. The results were expressed as mean ± standard deviations (SD). An analysis of the variance among different groups was conducted by Duncan’s multiple range test (*p* < 0.05) using SPSS 25.0 software (IBM, Chicago, IL, USA). All figures were drawn by Origin 9 software (OriginLab, Northampton, MA, USA).

## 3. Results and Discussion

### 3.1. MW Distribution of F30, F30-10, and F10

As a macromolecule, a protein can be degraded into MW-reduced peptides under enzymolysis, leading to the formation of MW-reduced and functionality-modified fragments. Therefore, the enzymatically hydrolyzed protein fractions can be characterized by their MW distribution. As exhibited in Figure 1A, four peaks were observed for F30, and three peaks were observed for F30-10 and F10. Obviously, peak 1 of F30 disappeared after further ultrafiltration, indicating that F30 possessed a portion with high MW. For clearly elucidating the MW distribution of these fractions, the peak MW (M_p_), relative contents (%) of each peak, and polydispersity (M_w_/M_n_) of peak 2 were listed in Appendix A. Compared with F30, F30-10 and F10 exhibited a gradually decreased relative content of peak 2 and a progressively increased relative content of peak 4. These results indicated that, during further ultrafiltration, the fractions with high MW for F30 were trapped, and the fractions with small MW were separated through ultrafiltration membrane. However, some hydrolysates with small MW could be easily connected with hydrolysates with large MW through hydrophobic and electrostatic interactions, leading to the appearance of peaks with small MW in the three fractions [14]. Moreover, some physical factors, such as temperature, pressure, and centrifugation, can cause the formation of aggregates with large MW among protein hydrolysates with small MW via hydrophobic interactions and van der Waals forces in the unstable hydrolysate system [15]. Therefore, the small MW hydrolysate could not be separated from the large MW hydrolysates in the neutral water and a lot of small MW hydrolysates might still be contained in F30. In addition, the polydispersity (peak 2) of F30 was the highest, followed by that of F30-10 and F10, indicating that the MW distribution of the obtained fractions became narrower with the increased amount of ultrafiltration. The GPC analysis of ultrafiltered fractions demonstrated that F30, F30-10, and F10 possessed a gradually decreased MW distribution.

Figure 1B depicts the electrophoretogram of SPI, SPIH, and ultrafiltered fractions. As exhibited in lane 2, the SPI was confirmed to be composed of the 7S protein and 11S protein. The 7S protein mainly consists of three subunits, including α’ (72 kDa), α (68 kDa), and β (52 kDa). Furthermore, the 11S protein includes subunits of A3 (45 kDa), acidic chains (35 kDa), and basic chains (20 kDa). These results agree with those reported in our previous study [16]. After being enzymatically hydrolyzed, as shown in lane 3, the intensity of bands with MW > 30 kDa apparently weakened or some of these bands disappeared; meanwhile, the diffused bands with MW < 10 kDa emerged. The band distribution of F30 exhibited a similar profile, compared with that of SPIH, suggesting that small MW hydrolysates were contained in F30 (also shown in Appendix A). The small MW hydrolysates could be separated from other high MW hydrolysates in SDS and β-mercaptoethanol-containing buffers, causing a continuous band in the area with MW < 10 kDa. This result agrees with that reported by Ahmadifard et al. [17]. Compared with the band profile of F30, the intensity of bands in the MW range of 17~35 kDa dramatically reduced for F30-10 and these bands nearly disappeared in the electrophoretogram of F10. Moreover, the intensity of protein bands for F10 was evidently reduced, which might be related to the easy running off the gel of the small MW hydrolysates contained in F10. Therefore, SDS-PAGE analysis also demonstrated that F30, F30-10, and F10 possessed different protein compositions.

### 3.2. Characterization of F30, F30-10, or F10-Dextran Conjugates

The characterization of F30, F30-10, or F10-dextran conjugates was evaluated by measuring the absorbance values at 304 nm (A_304_) and 420 nm (A_420_), which represent the content of Amadori compounds and melanoidins formed in the initial or final stages of the Maillard reaction, respectively. As shown in Figure 1C, as the incubation time increased, the A_304_ for all conjugates gradually increased, indicating that covalent conjugation indeed occurred between F30, F30-10, or F10 and dextran. The A_420_ exhibited the same change tendency. In the final stage of the Maillard reaction, the advanced products such as melanoidins with a deep color and unacceptable aroma were formed, resulting in an increase in the degree of browning. In addition, the F30-dextran conjugate showed higher A_304_ and A_420_ nm than the F30-10-dextran and F10-dextran conjugates, indicating that the high MW SPIH tended to undergo the Maillard reaction to a great extent. Li et al. [18] investigated the effects of the hydrolysis degree on the functionalities of the rice protein hydrolysate-saccharide conjugates and the results showed that the high hydrolysis degree caused a movement of lysine to a more hydrophobic region, making it difficult for the reactive −NH_2_ to conjugate with saccharide. Therefore, the relatively lesser A_304_ and A_420_ found in the F30-10- or F10-dextran conjugate was probably related to the more exposed hydrophobic groups in these relatively small MW hydrolysates.

To further understand the covalent conjugation between SPI hydrolysate fractions and dextran, SDS-PAGE was applied to estimate the MW distribution of the F30, F30-10, or F10-dextran conjugates during incubation (Figure 2). As shown in Figure 2b,d,f, the protein bands of F30, F30-10, and F10 were obviously unchanged during the incubation at 60 °C for 24 h, indicating that the heat treatment did not change the subunit composition of SPIH. Furthermore, all F30, F30-10, or F10-dextran mixtures (lane 1 in Figure 2a,c,e) exhibited nearly the same electrophoresis patterns compared with F30, F30-10, and F10, suggesting that there was only a simple mixing between F30, F30-10, and F10 and dextran before heating.

However, all original bands of the heated F30, F30-10, or F10-dextran mixtures gradually disappeared with the increasing incubation time. A broad band marked by the red-dotted box was observed at the top of the separating gel, suggesting the occurrence of covalent conjugation between F30, F30-10, or F10 and dextran. Meanwhile, this broad band could not move down through the gap of the gel, owing to the decreased electrophoretic mobility, suggesting that F30, F30-10, or F10-dextran conjugates with high MW were generated due to the Maillard reaction. These results agreed with those reported in a previous study, in which SPI was initially enzymatically hydrolyzed and then glycosylated with maltodextrin using a wet-heating method [19].

### 3.3. Structural Properties of F30, F30-10, or F10-Dextran Conjugates

#### 3.3.1. UV Absorption Intensity

The side chain groups of tryptophan and tyrosine residue always confer a detectable ultraviolet absorption spectrum to the proteins; therefore, the conformational changes could be assessed by the differences in the ultraviolet absorption of proteins [20]. As illustrated in Figure 3A, all samples showed a characteristic absorption peak at 270–400 nm, and the wavelength with maximum absorption peak (λ_max_) was around 280 nm. After hydrolyzing by neutrase, the SPI molecules unfolded and degraded; thus, more relatively small MW peptides emerged, leading to an increase in the UV absorption intensity. This finding also coincides with the result reported by Liu et al. [21], who revealed that the UV absorption intensity of ovalbumin at 280 nm was significantly increased after hydrolysis by Alcalase. However, the UV absorption intensity of F30, F30-10, and F10 at 280 nm decreased to be less than that of the native SPI. This might be related to the blocking effect of the side chain groups of tryptophan and tyrosine residues by the possible aggregation between the small MW hydrolysates, which could be reflected by the uneven MW distribution of F30, F30-10, and F10 mentioned above.

For mixed systems of F30, F30-10, or F10 and dextran (Figure 3B–D), the UV absorption intensity of glycosylated products varied significantly in a range of 270–400 nm during incubation at 60 °C. The absorption intensity of the F30-dextran and F30-10-dextran conjugates at 280 nm increased as the incubation time increased; however, the F30-dextran conjugate exhibited a greater increase in amplitude. On the contrary, with the increasing incubation time, the absorption intensity of the F10-dextran conjugates at 280 nm exhibited a trend of first decreasing and then increasing. These changes demonstrated the formation of the Maillard reaction products [21]. Generally, the grafting of polysaccharide chains promotes the protein structure to unfold, which is conducive to the exposure of aromatic amino acids, such as tyrosine and tryptophan, and increases the UV absorption intensity of glycosylated products [22]. In this study, the UV absorption intensity of F30, F30-10, and F10 showed different changes after modification by glycosylation. This might be related to the relatively high glycosylation reactivity of F30 mentioned above; thus, the amount of grafted polysaccharide chains and the influence on protein structure were greater than that of F30-10 or F10. Moreover, F10 has the smallest MW and the most exposed hydrophobic groups. Therefore, accompanying the grafting of polysaccharide chains on protein molecules, the aggregation among F10 molecules would also occur through hydrophobic interaction to cause a weak absorption of tryptophan and tyrosine residues.

#### 3.3.2. Tertiary Structure

As exhibited in Figure 3E, when the excitation wavelength was 290 nm, SPI showed a maximum fluorescence intensity at 336 nm, which was the typical intrinsic fluorescent absorption of tryptophan residues in a hydrophobic environment. Compared with SPI, the maximum fluorescence intensity of SPIH at 336 nm decreased, but a new absorption peak at 306 nm was formed. The fluorescence absorption of SPIH at 306 nm was mainly related to tyrosine and phenylalanine residues, indicating that enzymolysis caused SPI structures to unfold [23]. The fluorescence intensity of F30, F30-10, or F10 increased obviously after ultrafiltration. The fluorescence spectrum of F30 is dominated by the absorption peak around 338 nm, while the fluorescence spectrum of F30-10 or F10 is dominated by the absorption peak around 302 nm. These phenomena suggest that the smaller the MW of SPIH is, the more the fluorescence absorption of tyrosine and phenylalanine residues can be observed at 306 nm.

As shown in Figure 3F–H, the fluorescence intensity decreased obviously when F30, F30-10, and F10 covalently bound with dextran; moreover, F30-dextran conjugates exhibited the most dramatic decrease in fluorescence intensity. The polysaccharide, which is covalently bound to polypeptide, would screen the chromophore of the tryptophan residue, resulting in a significant decrease in the maximum fluorescence intensity. In addition, this was also likely associated with a conformational change in the proteins induced by glycosylation. This result is consistent with that reported by Spotti et al. [24], who found that the fluorescence intensity of WPIs decreased significantly after glycosylation with glucan. Moreover, due to the relatively high reactivity of F30-dextran conjugates, the steric hindrance effect of polysaccharides was enhanced, leading to an enhanced shielding effect on tryptophan residues and, consequently, the maximum decrease in fluorescence intensity [25]. During incubation, the λ_max_ of F30-dextran conjugates shifted towards a long-wavelength direction; thus, a red shift occurred. However, the λ_max_ of F30-10-dextran and F10-dextran conjugates exhibited no changes. Generally, a red shift of λ_max_ occurs when tryptophan residues are exposed to strong polar conditions; therefore, this phenomenon is usually used to indicate the unfolding of the protein structure [26]. Thus, this result suggests that the grafting of dextran chains increased the polarity of tryptophan chromogenic group microenvironment of F30. The more pronounced this phenomenon is, the looser the tertiary structure and the higher the molecule flexibility of reactant protein are [25].

#### 3.3.3. FTIR Spectroscopy

Generally, during glycosylation, an amino group of proteins tends to form covalent bonds with carbonyl group of polysaccharides; thus, the primary structures (mainly C-N and N-H bonds) of the protein are modified. The infrared spectra of all studied samples are shown in Figure 4. As depicted in Figure 4A, SPI exhibited three characteristic bands at 1659 cm^−1^ (Amide I, C=O stretching), 1540 cm^−1^ (Amide II, C-N stretching and N-H bending), and 1237–1446 cm^−1^ (Amide III, C-N stretching and N-H bending vibrations) [27]. However, the absorption intensity of SPIH at Amide II weakened and further decreased after ultrafiltration. The same decrease in the absorption intensity of Amide II was reported in Flavourzyme-hydrolyzed peanut protein isolate due to the loss of some secondary structure and the reduced protein content after hydrolysis [28]. In addition, the changes in the absorption peak at 3000–2800 cm^−1^ (C-H symmetric and asymmetric stretching vibration) could be used to characterize the change in the hydrophobic regions of proteins. The absorption peak of SPI at 2929 cm^−1^ was related to the CH_2_ stretching vibration, which usually occurred in the aliphatic side chain of proteins [29]. Thus, the significant decrease in the absorption peak of SPIH at 2929 cm^−1^ indicated that enzymolysis could change the microenvironment of the alkyl chain and open the hydrophobic region of SPI. However, the absorption intensity of SPIH at 1082 and 540 cm^−1^ was obviously higher than the SPI. Moreover, new absorption bands appeared at 1164, 1077, 950, and 860 cm^−1^ in the spectra of F30, F30-10, and F10, and the absorption intensity of these peaks gradually increased with the decreasing MW. These results indicated that the structure of SPI changed significantly after enzymolysis.

For dextran, a series of overlapping peaks (1155–1015 cm^−1^) is mainly ascribed to the stretching vibration of C-C and C-O bonds and the bending mode of C-H bonds (Figure 4A); the absorption peak around 915 cm^−1^ confirmed the presence of α-glycosidic bonds of polysaccharide [30]. As exhibited in Figure 4B–D, for F30, F30-10, or F10-dextran conjugates, the intensity of the absorption peaks at Amide I and Amide III increased, compared with those of their mixtures. For instance, the intensity of the absorption peak (1360–1310 cm^−1^) induced by C-N stretching vibration in SPIH-dextran conjugate was nearly increased as the reaction time increased (exhibited in Figure 4B–D) [31]. This discrepancy might be related to an absorption enhancement caused by the formation of Schiff base, carbonyl-amine compounds, and pyrazine compounds during glycosylation reaction [32]. It was worth noting that within the wavelength of 1200–960 cm^−1^, the absorption intensity of conjugates was higher than that of F30, F30-10, and F10. This was likely attributed to C-O-C stretching vibrations of dextran. The enhanced absorption intensity at 3423 cm^−1^ of conjugates was likely caused by the stretching vibration absorption of free OH groups provided by the grafted dextran and related to the presence of a new N-H bond during glycosylation. Furthermore, the absorption intensity at 2930, 2873, 861, and 540 cm^−1^ for conjugates was also enhanced, which might be due to the C-H antisymmetric stretching vibration of CH_3_ and CH_2_ groups and demonstrates the occurrence of glycosylation [32]. Therefore, the glycosylation between SPIH and dextran generated a new functional group, which caused the vibration of protein side chains and enhanced the absorption intensity of the relevant peaks.

#### 3.3.4. Thermal Stability

As exhibited in Figure 5, all samples exhibited endothermic peaks of water evaporation and decomposition in proteins over a wide temperature range (20–250 °C). Generally, the curve area is used to indicate enthalpy change (Δ*H*), which is contributed by the energy for overcoming noncovalent interactions during protein denaturation and the peak temperature (*T*_p_) reflects the denaturation temperature of proteins. It could be seen that SPI had an obvious endothermic peak at 123.66 °C and Δ*H* was 163.10 J/g. However, after hydrolysis by Neutrase, *T*_p_ of SPIH increased to 136.00 °C, but Δ*H* decreased to 158.50 J/g. These changes suggest that an appropriate enzymolysis could enhance the thermal stability of proteins. Wang et al. [33] also reported an increased denaturation temperature of wheat gluten after Alcalase-based partial hydrolysis; this phenomenon might be ascribed to the change in conformation of the protein, especially the hydrolysis-induced high proportion of hydrophobic regions. The significant decrease in Δ*H* might be related to the flexible or disordered structure of proteins induced by the loss of tertiary structure of proteins by protease hydrolysis [34].

In addition, it was found that *T*_p_ of F30 (124.80 °C) was greater than that of F30-10 (121.65 °C) and F10 (122.37 °C), while *T*_p_ of these SPI fractions were less than that of SPIH. This result suggests that the thermal stability of F30 is stronger than that of F30-10 and F10, but weaker than that of SPIH. Furthermore, new endothermic peaks appeared at 60.91 °C and 45.36 °C for F30-10, while for F10, this peak temperature was further reduced to 54.22 °C and 44.62 °C. On the contrary, Δ*H* of each endothermic peak increased with the decreasing fraction MW. The endothermic peaks in the DSC curve of proteins are usually related to the fracture of hydrogen bonds [35]. F30-10 and F10 were mainly composed of polypeptide chains with low MW, compared with F30. Therefore, hydrogen bonds which maintain the spatial structure of proteins broken down to intensify the aggregation of polypeptide chains, and then cause a decrease in the thermal stability of SPIH fractions.

As shown in Figure 5B–D, *T*_p_ of F30, F30-10, or F10-dextran conjugates was greater than that of their corresponding mixtures, and Δ*H* was much less than that of the mixtures, indicating that glycosylation indeed occurred between these SPIH fractions and dextran [36]. Moreover, this modification enhanced the thermal stability of SPIH fractions. F30-10 and F10 showed a greater reduction in Δ*H* after glycosylation, compared with F30. As explained previously, the grafting of polysaccharide chains not only changed the hydrophilicity/hydrophobicity balance by increasing the net negative charge and hydroxyl groups on the surface of native protein, but also enhanced the steric hindrance of conjugates [37]. Thus, the glycosylated products had stable spatial structures and were not easy to aggregate when heated. As reported by Liu et al. [38], compared with the native peanut conarachin and arachin, the peanut conarachin or arachin-dextran conjugate showed a higher denaturation temperature and a lower enthalpy change. After grafting of dextran chains, the chemical bonds in F30, F30-10, and F10 molecules, such as hydrogen bonds, hydrophobic bonds, disulfide bonds, and electrostatic interactions, changed significantly, thus, further enhancing the structural integrity, connectivity, and the thermal stability of the conjugates [6].

### 3.4. Antioxidant Activities

#### 3.4.1. DPPH Radical Scavenging Activity

As shown in Figure 6A, the DPPH radical scavenging activity of all samples was higher than that of V_C_ (positive control, 49.73 ± 3.35%). The antioxidant activities of native SPI might be related to the type, amount, amino acid composition, and surface hydrophobicity [39]. However, at the same concentration, the DPPH radical scavenging activity of SPIH (78.31 ± 2.08%) was greater than that of SPI (65.81 ± 4.68%). Generally, protein hydrolysates could provide more electron donors that can react with free radicals to convert them to a stable product, thus, terminating the free radical chain reaction [8]. After the ultrafiltration of SPIH, the DPPH radical scavenging activity of F30, F30-10, and F10 was reduced to 71.42 ± 0.12%, 64.28 ± 0.62%, and 60.31 ± 0.26%, respectively. This might be related to their different structures and physicochemical properties. For instance, the DPPH radical scavenging ability of protein hydrolysates closely correlated with their hydrophobic amino acids and surface hydrophobicity [40]. In general, the lower the surface hydrophobicity is, the stronger the free radical scavenging ability exhibits. Therefore, with the decrease in MW of SPIH, more hydrophobic groups are exposed, leading to an increase in the surface hydrophobicity and, finally, a reduced DPPH radical scavenging ability.

Nevertheless, the DPPH radical scavenging activity of F30, F30-10, and F10 significantly enhanced after glycosylating with dextran (Figure 6B), suggesting that the grafting of dextran improved the antioxidant activities of SPIH. A similar result was also reported when neutrase-based SPIH was glycosylated with maltodextrin using a wet-heating method [19]. In addition, the DPPH radical scavenging activity of F30/F30-10/F10-dextran conjugates gradually increased with the increasing incubation time; however, the less the MW of the SPIH fraction was, the more significant increase in amplitude was exhibited. It could be also seen that the DPPH radical scavenging activity of F30-dextran conjugate was significantly greater than that of the F30-10 or F10-dextran conjugate for less incubation time (<4 h). However, when the incubation time was extended, F30-10 or F10-dextran conjugate showed a higher DPPH radical scavenging activity. These discrepancies probably correlated with different protein compositions and followed different glycosylation degrees among F30, F30-10, and F10. Therefore, the improvement of the DPPH radical scavenging activity of SPIH by grafting with dextran chains might be closely related to the molecular size of SPIH fractions.

#### 3.4.2. •OH Scavenging Activity

As shown in Figure 6C, SPI and its hydrolysates exhibited higher •OH scavenging activity compared with V_C_ (7.73 ± 1.80%). More importantly, the •OH scavenging activity of SPIH (28.78 ± 0.45%) was significantly greater than that of SPI (20.30 ± 0.71%, *p* < 0.05). Furthermore, the OH scavenging activity of F30, F30-10, and F10 was 33.00 ± 3.12%, 38.19 ± 2.36%, and 36.38 ± 0.74%, respectively, suggesting that the SPIH with a low MW (F30-10 and F10) had a stronger •OH scavenging activity than that with a high MW (F30). The chickpea protein hydrolysate with a low MW also exhibited strong •OH scavenging activity due to the high concentration of hydrophobic regions containing hydrophobic amino acids [41]. Compared with F30, F30-10, and F10, as exhibited in Figure 6D, the conjugates exhibited a higher •OH scavenging activity; moreover, with the increasing reaction degree, the •OH scavenging activity of glycosylated products was slightly enhanced. In addition, during the entire reaction process, F30-10-dextran and F10-dextran conjugate exhibited higher •OH scavenging activity than the F30-dextran conjugate, which also indicated that the SPIH fraction with a small molecular size exhibited relatively high •OH scavenging activity.

#### 3.4.3. Ferrous Reducing Power

As shown in Figure 6E, compared with the native SPI, SPIH had a weaker ferrous reducing power. After the ultrafiltration of SPIH, the ferrous reducing power of F30, F30-10, and F10 was significantly enhanced to 0.115 ± 0.001, 0.103 ± 0.002, and 0.105 ± 0.001, respectively, suggesting that the SPIH with a greater MW exhibited higher ferrous reducing powers. This was the same as the trend of the DPPH radical scavenging ability. Moreover, with the increased incubation time, the F30-dextran conjugates exhibited higher ferrous reducing power than the F30-10- or F10-dextran conjugates; however, after 6 h of incubation, the increase in the ferrous reducing power of the F30-dextran conjugate changed to be slow.

Combined with the results of the DPPH radical or •OH scavenging assays, SPIH exhibited a stronger DPPH radical and •OH scavenging activity and weaker ferrous reducing power compared with SPI. However, F30, F30-10, and F10 showed higher DPPH radical or •OH scavenging activity and ferrous reducing power after glycosylation with dextran. With the proceeding of glycosylation, antioxidant activities were significantly increased. Many studies have shown that the Maillard reaction products have an antioxidant activity [42]. For instance, hydroxyl groups or reduced pyrrole and furan groups in the Maillard reaction products, as well as some antioxidant components produced in the intermediate or advanced stages of the reaction, such as macromolecular melanoid, reduced ketones, volatile heterocyclic compounds, and various intermediates, can play as hydrogen donors in scavenging free radicals and thus have strong a free radical scavenging ability. Meanwhile, these compounds can also play a reducing role and are good electron donors to supply electrons to reduce Fe^3+^ to Fe^2+^, resulting in a higher ferrous reducing power of conjugates [43]. Compared with F30-10- or F10-dextran conjugates, the antioxidant activities of F30-dextran conjugates increased slowly at the late reaction period (12–24 h). This might be related to the relatively high degree of grafting when F30 glycosylated with dextran. Accompanying the occurrence of carbonyl-amino condensation, some intermediate products with antioxidant activities might degrade, due to the possible advanced Maillard reaction, which would affect the antioxidant activities of conjugates.

## 4. Conclusions

SPI was successfully hydrolyzed by neutrase following ultrafiltration and glycosylation with dextran using a dry-heating method to obtain the thermal stability- and antioxidant activities-enhanced conjugates. F30 with high MW was proven to be more susceptible to conjugates with dextran compared with F30-10 or F10. The grafting of dextran further promoted the unfolding of protein conformation to obtain a conjugate with loose spatial structure and high molecular flexibility, which is conducive to the formation of conjugates with improved thermal stability and antioxidant activities. This change was more pronounced for F30 than the other two SPIH fractions. Moreover, the F30-dextran conjugate exhibited a higher DPPH radical scavenging activity and ferrous reducing power than F30-10 or F10-dextran conjugates at incubation time <4 or 6 h, respectively; however, F10-dextran conjugate showed the highest •OH scavenging activity during incubation. Therefore, the results provide a feasible strategy for the further exploitation of SPI resources via the development and application of SPIH fractions (F30, F30-10, and F10) and their glycosylated products as effective antioxidants in functional foods.

## Figures and Tables

**Figure 1 antioxidants-12-00430-f001:**
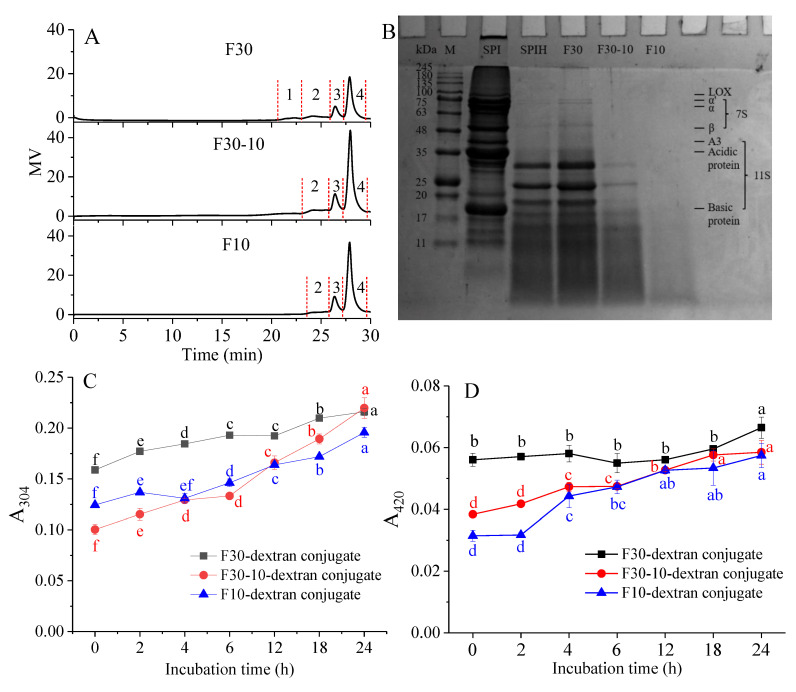
MW distribution eluted by GPC (**A**) and protein composition (**B**) of the F30, F30-10, and F10 ultrafiltrated from the SPIH and changes in amount of Amadori compounds ((**C**), A_304_) and browning degree ((**D**), A_420_) of F30-, F30-10, or F10-dextran conjugates incubated for 0, 2, 4, 6, 12, 18, and 24 h (Peak 1–4 in Figure 1A stand for the successively eluted partitions of SPIH using GPC; GPC, gel permeation chromatography; M, protein marker; SPI, soybean protein isolate; SPIH, SPI hydrolysate; different letters for the same color represent the significant difference, *p* < 0.05).

**Figure 2 antioxidants-12-00430-f002:**
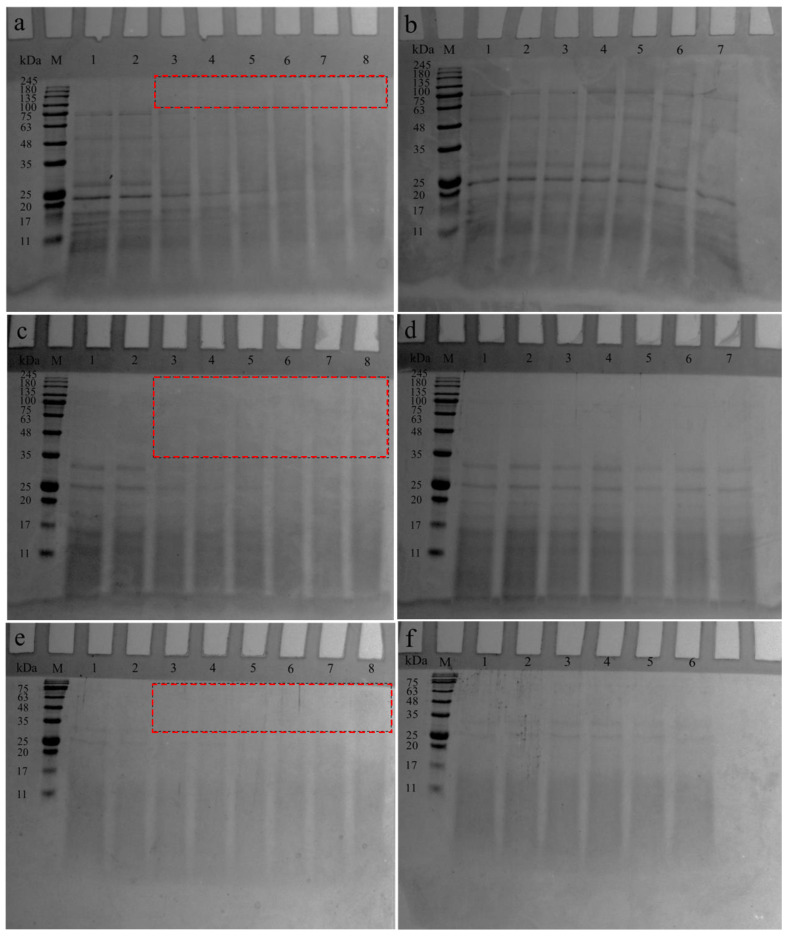
SDS-PAGE patterns of F30-dextran conjugate (**a**), F30 (**b**), F30-10-dextran conjugate (**c**), F30-10 (**d**), F10-dextran conjugate (**e**), and F10 (**f**) incubated at 60 °C for 0–24 h (M, Marker; for graph a, c, and e, lane 1 represents the F30, F30-10, or F10; lanes 2–8 represent the conjugates incubated for 0, 2, 4, 6, 12, 18, and 24 h; for graph (**b**) and (**d**), lanes 1–7 represent the F30 or F30-10 incubated for 0, 2, 4, 6, 12, 18, and 24 h; for graph (**f**), lanes 1–6 represent the F10 incubated for 2, 4, 6, 12, 18, and 24 h; the red-dotted box indicates the possible F30, F30-10, or F10-dextran conjugate formed during glycosylation).

**Figure 3 antioxidants-12-00430-f003:**
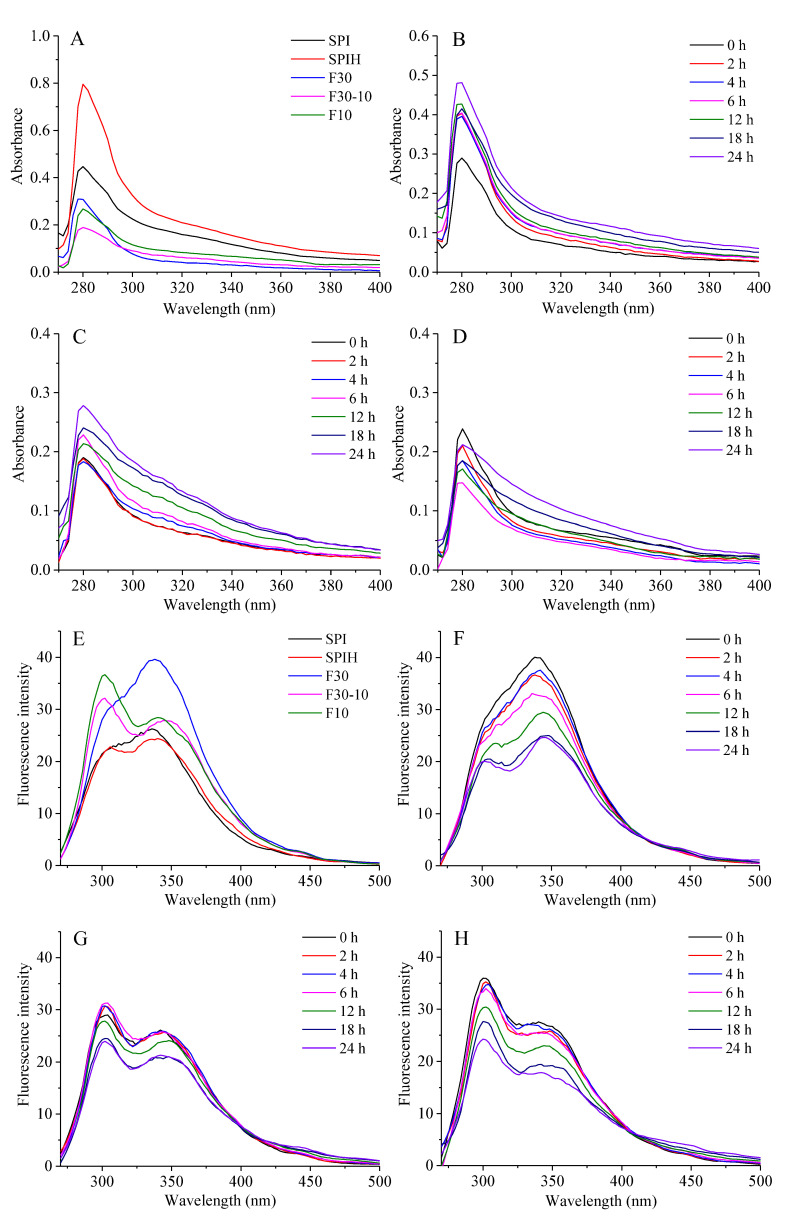
Ultraviolet spectra (**A**–**D**) and fluorescence spectra (**E**–**H**) of SPI, SPIH, F30, F30-10, and F10 (**A**,**E**), and F30-dextran (**B**,**F**), F30-10-dextran (**C**,**G**), and F10-dextran (**D**,**H**) conjugates incubated for 0, 2, 4, 6, 12, 18, and 24 h.

**Figure 4 antioxidants-12-00430-f004:**
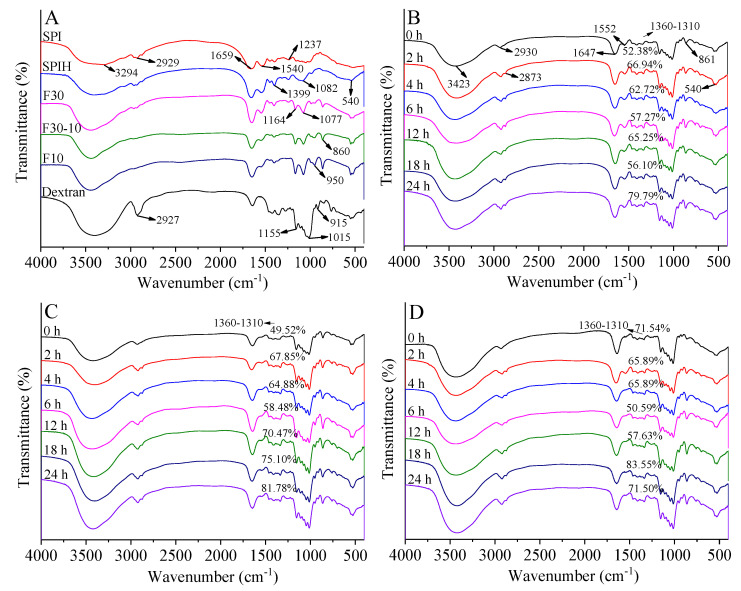
FTIR spectra of SPI, SPIH, F30, F30-10, and F10 (**A**); F30-dextran (**B**); F30-10-dextran (**C**); and F10-dextran (**D**) conjugates incubated for 0, 2, 4, 6, 12, 18, and 24 h.

**Figure 5 antioxidants-12-00430-f005:**
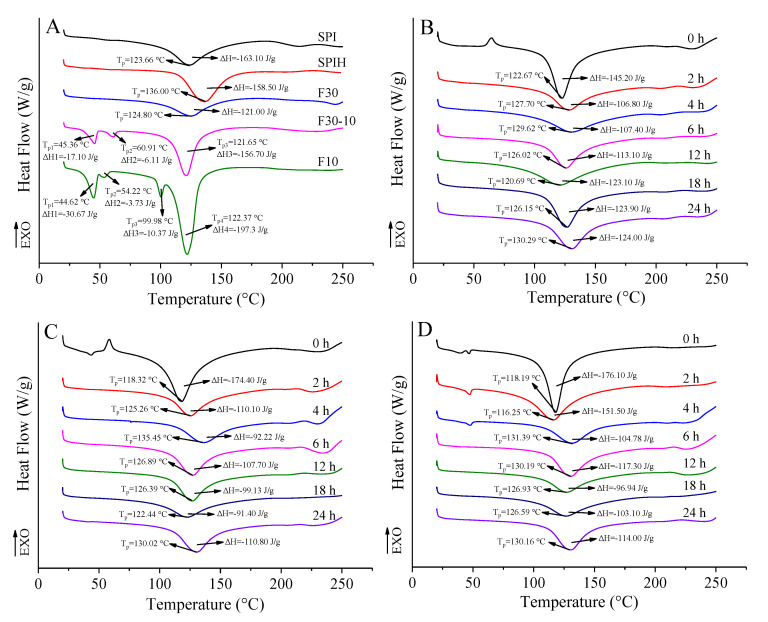
Thermograms of SPI, SPIH, F30, F30-10, and F10 (**A**); F30-dextran (**B**); F30-10-dextran (**C**); and F10-dextran (**D**) conjugates incubated for 0, 2, 4, 6, 12, 18, and 24 h.

**Figure 6 antioxidants-12-00430-f006:**
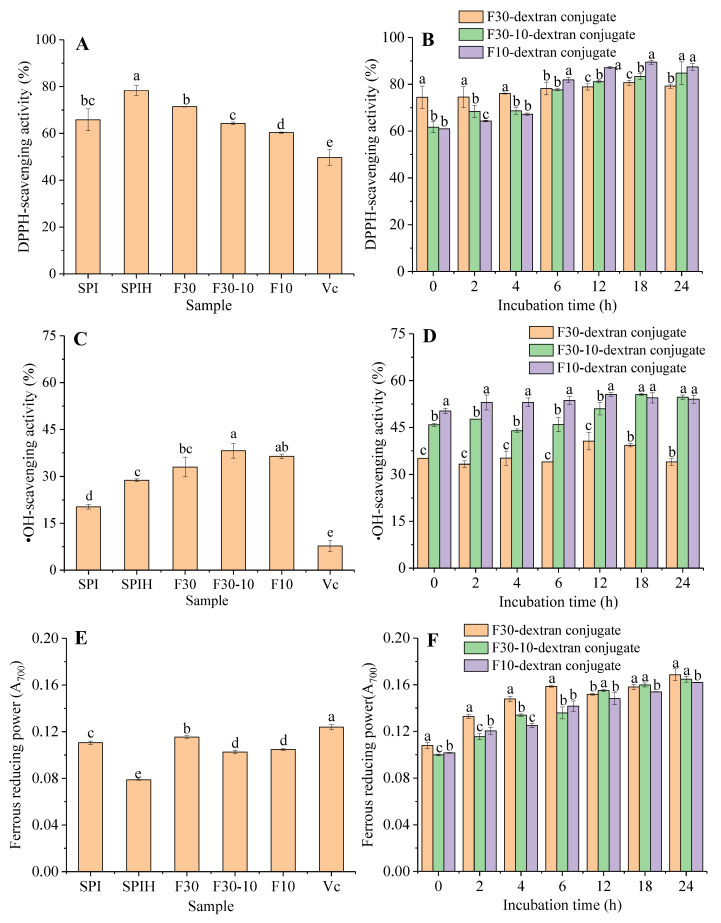
Antioxidant activities of the SPI, SPIH, F30, F30-10, F10, and V_C_ (**A**,**C**,**E**) and F30/F30-10/F10-dextran conjugates (**B**,**D**,**F**) incubated for 0, 2, 4, 6, 12, 18, and 24 h. (Different letters for conjugates incubated at the same time represent the significant difference, *p* < 0.05.)

## Data Availability

The data presented in this study are available in the article and Appendix A.

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
