# Peer review of "Effects of Sequential Enzymolysis and Glycosylation on the Structural Properties and Antioxidant Activity of Soybean Protein Isolate"

_antioxidants, 2023, doi:10.3390/antiox12020430_

Round 1

Reviewer 1 Report

Title: Effects of sequential enzymolysis and glycosylation on the structural properties and antioxidant activity of soybean protein isolate.

It is a good article, clear and carefully prepared. The motivation for the investigation of such a system is very well presented as well as the whole manuscript is written precisely. It may be even said that it is too much for one manuscript. Good and clear conclusions.

I recommend the publication of the manuscript in Antioxidants after minor revision.

English language should be improved.

Line 85: Was pH stable after adjusting using 0.1 M NaOH?

Line 155: Omitted numbering.

Figure 1 A Should be MW

Figure 2 Poor quality of images. Should be improved

Author Response

Response to Reviewer #1

Dear reviewer,

Thank you so much for your valuable comments and suggestions, which have been quite helpful to improve our manuscript. Your efforts in the review process of this manuscript are greatly appreciated. We have been replying to your comments as described below.

1) It is a good article, clear and carefully prepared. The motivation for the investigation of such a system is very well presented as well as the whole manuscript is written precisely. It may be even said that it is too much for one manuscript. Good and clear conclusions. I recommend the publication of the manuscript in Antioxidants after minor revision.

Thank you for your positive consideration for this manuscript.

2) English language should be improved.

English writing is substantially revised now.

3) Line 85: Was pH stable after adjusting using 0.1 M NaOH?

Yes, the pH was stable after adjusting using 0.1 M NaOH or HCl. pH 7.0 was maintained to promote the effective enzymolysis by Neutrase.

4) Line 155: Omitted numbering.

Thank you for this detailed comment. The omitted numbering of subtitle is revised now.

5) Figure 1 A Should be MW.

Authors carefully rechecked the title of y-axis of Figure 1A and the MV was correct. MV indicates the difference in refractive index between the sample and the solvent when gel permeation chromatography tandem differential refractive index detector is applied to measure the molecular weight distribution of polymers.

6) Figure 2 Poor quality of images. Should be improved.

The electrophoretogram is replaced by a version with high resolution now.

Reviewer 2 Report

The work carried out by Zhang and collaborators aimed to determine the effects of sequential enzymatic hydrolysis and glycosylation of SPI on the structural and antioxidant properties of the proteins.

The work is interesting, however in its current state is not acceptable for publication and therefore I must recommend that the work is rejected.

Major comments:

1) Despite that the work states in the title that aims to determine the effects of proteolytic digestion followed by glycosylation of SPI on its structural properties there is a lack of robust experiments to determine structural modifications on proteins (i.e. NMR, LC-MS, X-ray).

2) DPPH radical scavenging activity assay is an accepted methodology to investigate antioxidant properties of molecules, however, due to the chemical nature of the compound this can interact with proteins which may lead to errors in the interpretation of results. Why the authors did not performed any complementary experiments? I wonder whether dextran was removed or not before carrying out experiments to determine the *OH-scavenging activity of the samples. Dextran polymers are poly-alcohols therefore these can also react with *OH. There is a lack of control experiments to determine a possible contribution of dextran.

Minor comments:

1) The grammar of this work needs to be revised by a native English speaker.

2) Did the authors consider the possible role of advanced glycation endproducts generated after long incubation times? These may be generated and there is strong evidence linking formation/consumption of these with detrimental health consequences.

3) The authors conclude that combination of enzymatic digestion and glycosylation confer enhanced functionalities to SPI. However, this is something still under debate. I would try to omit such a statement.

4) I would strongly encourage that the authors perform size exclusion chromatography or a similar technique together with SDS-PAGE analyses shown in Fig1. The resolution of the gels is clearly not the indicate to resolve proteins/peptides in the low molecular weight range, which is the most relevant in this work.

5) I am not convinced by Fig2. Where SDS-PAGE gels stained with Coomassie or using silver staining?

6) fluorescence spectra of proteins and peptides are clearly not comparable as the first possess microenvironments whilst peptides, depending on their length and formation of secondary structures, may or may not have these.  

Author Response

Response to Reviewer #2

Dear reviewer,

Thank you so much for your valuable comments and suggestions, which have been quite helpful to improve our manuscript. Your efforts in the review process of this manuscript are greatly appreciated. We have been replying to your comments as described below.

1) The work is interesting, however in its current state is not acceptable for publication and therefore I must recommend that the work is rejected.

Thank you for your comment. The manuscript is substantially revised according to your valuable revision.

2) Despite that the work states in the title that aims to determine the effects of proteolytic digestion followed by glycosylation of SPI on its structural properties there is a lack of robust experiments to determine structural modifications on proteins (i.e. NMR, LC-MS, X-ray)..

3) Indeed, the reviewer provided a pertinent comment regarding the characterization of hydrolysates and the followed conjugates using structural analysis techniques. Authors agree this study thought which a robust experimental design for a further research of the combined modification of native or processed proteins. In this research, authors provided the enzymatic results to a certain degree using gel permeation chromatography analysis and the glycosylation results using the analyses of Amadori compounds amount and browning degree and changes in protein subunit composition. These techniques or indexes are also commonly used methods for the confirmation of the production of hydrolysates and protein-polysaccharide conjugates.

3) DPPH radical scavenging activity assay is an accepted methodology to investigate antioxidant properties of molecules, however, due to the chemical nature of the compound this can interact with proteins which may lead to errors in the interpretation of results. Why the authors did not performed any complementary experiments? I wonder whether dextran was removed or not before carrying out experiments to determine the *OH-scavenging activity of the samples. Dextran polymers are poly-alcohols therefore these can also react with *OH. There is a lack of control experiments to determine a possible contribution of dextran.

Thank you for this innovative comment. Authors only considered the controls from the protein’ point of view and did not consider the impact of free dextran on the DPPH-radical scavenging activity when the antioxidant properties of soybean protein isolate hydrolysate-dextran conjugate was measured. The free dextran problem is related to the overmuch dextran that might be not involved in glycosylation. The initial research objective was the investigation of the effects of combined enzymatic hydrolysis and following glycosylation on the structural properties and antioxidant activities of the modified products. Theoretically, after these modifications (especially enzymolysis), the structure of SPI might be greatly changed, which can be reflected by the analyses of protein subunit composition, UV spectroscopy, conformation, and FTIR spectroscopy.

4) The grammar of this work needs to be revised by a native English speaker.

Thank you for your advice. Authors have invited a professional writer to revise the English writing of this manuscript.

5) Did the authors consider the possible role of advanced glycation endproducts generated after long incubation times? These may be generated and there is strong evidence linking formation/consumption of these with detrimental health consequences.

Theoretically, glycosylation is just the initial stage of the Maillard reaction so that no advanced reaction endproducts are generated under the mild reaction conditions. The production of advanced reaction endproducts not only causes health concern but also goes against the modification of native proteins. Therefore, relative mild reaction temperature and relative humid (60 °C and 79%) are commonly selected to prepare protein-polysaccharide conjugates (Akhtar M. and Ding R. Covalently cross-linked proteins & polysaccharides: Formation, characterisation and potential applications. Current Opinion in Colloid & Interface Science, 2017, 28, 31-36; de Oliveira, F.C., et al. Food protein-polysaccharide conjugates obtained via the maillard reaction: A Review. Critical Reviews in Food Science and Nutrition, 2016, 56, 1108-1125; Zhang et al., Protein glycosylation: a promising way to modify the functional properties and extend the application in food system. Critical Reviews in Food Science and Nutrition, 2019, 59, 2506-2533). Moreover, lots of previous studies prepared protein-polysaccharide conjugates under this type of conditions using dry-heating method.

6) The authors conclude that combination of enzymatic digestion and glycosylation confer enhanced functionalities to SPI. However, this is something still under debate. I would try to omit such a statement.

Thank you for this meaningful comment. Authors concluded that the combination of limited enzymolysis and glycosylation could prepare SPI-based food ingredient with enhanced functionalities compared with native SPI, not the enhanced functionalities to or of SPI. However, the previous description was still revised based on this comment.

7) I would strongly encourage that the authors perform size exclusion chromatography or a similar technique together with SDS-PAGE analyses shown in Fig1. The resolution of the gels is clearly not the indicate to resolve proteins/peptides in the low molecular weight range, which is the most relevant in this work.

Thank you for this important advice. Indeed, SDS-PAGE analysis is not sufficient to indicate the protein composition after enzymolysis; therefore, authors conducted gel permeation chromatography analysis of SPI hydrolysates (the denoted F30, F30-10, and F10) and the results were exhibited in Fig. 1A and Table S1.

8) I am not convinced by Fig 2. Where SDS-PAGE gels stained with Coomassie or using silver staining?

As described in the section of 2.4, after separating, gels were stained with 0.25% (w/v) Coomassie Brilliant Blue R-250 for 50 min, and then de-stained using 25% methanol and 10% acetic acid for scanning using an image scanner. Compared with iodine staining, copper staining, potassium chloride, and silver staining, staining by Coomassie Brilliant Blue is a commonly used method to conduct the protein composition analysis. It is worth noting that although the silver staining exhibits a high sensitivity, it is a complex procedure and easily interfered by environmental stresses. So far, staining by Coomassie Brilliant Blue is extensively used to characterize the changes in subunit composition before and after glycosylation. As exhibited in Fig. 2, compared with controls (Fig. 2b, d, and f), some bands clearly disappear and dispersive bands at the top of separating gel emerge, indicating that covalent glycosylation occurs between SPI hydrolysates and dextran.

9) Fluorescence spectra of proteins and peptides are clearly not comparable as the first possess microenvironments whilst peptides, depending on their length and formation of secondary structures, may or may not have these.

Thank you for this detailed comment. Indeed, authors agree with the reviewer’s viewpoint which the peptides do not always possess microenvironments. In this study, the fluorescence spectra of SPI and its hydrolysates were determined to be used as controls, which benefits on the investigation of the covalent glycosylation between SPI hydrolysates and dextran. The difference in fluorescence spectra of SPI and its modified products is just the result of changes in structure.

Reviewer 3 Report

The study by Zhang et al., reported that antioxidant activity of the glycosylated soybean protein isolate hydrolysate. 

Authors used various methods to characterize their structures. However, it is difficult to understand relationship between structure and antioxidant activity.

Minor comments

Why is enzymatic hydrolysis necessary? Please add some reasons in the manuscript.

Why authors used dextran instead of dextrin. Please add some reasons in the manuscript.

Did you check the glycosylation of soybean protein isolate (without enzymatic hydrolysis)?

What is the percentage of glycosylation? Because the changes in the graphs of 1C and 1D do not seem to match.

The explanation of the dot should be included in the description of Figure 2.

I don't understand the difference within the dots in Figure 2. Please change to a easier figure to understand.

Please show the absorption intensity of newly formed C-N bonds by the Maillard reaction (Figure 4).

Authors mentioned endothermic peaks appeared at 60.91 °C and 45.36 °C for F30-10, while for F10, this peak temperature was further reduced to 54.22 °C and 44.62 °C. Those peaks are not found at 0 h in Fig. 5C D. Please explain what these differences are.

The authors should indicate the units of Ferrous reducing power.

In this paper, authors produced three different sizes of glycated protein hydrolysates. Are there any ways to utilize the characteristics of each SPHI?

Author Response

Response to Reviewer #3

Dear reviewer,

Thank you so much for your valuable comments and suggestions, which have been quite helpful to improve our manuscript. Your efforts in the review process of this manuscript are greatly appreciated. We have been replying to your comments as described below.

1) Authors used various methods to characterize their structures. However, it is difficult to understand relationship between structure and antioxidant activity.

Thank you for this valuable comment. Indeed, the original results and discussion did not directly indicate the relationship between structural properties and antioxidant activity. The part of structural properties aims to confirm the preparation of SPI hydrolysates and SPI-dextran conjugates and the tertiary conformation, typical infrared absorption groups, and thermal properties of obtained products, which provides fundamentals to the differences in antioxidant activities of SPI hydrolysate-dextran conjugates.

2) Why is enzymatic hydrolysis necessary? Please add some reasons in the manuscript.

Modification of protein functionalities via moderately change its structure is a commonly used method to obtain modified protein. So far, physical modification, chemical modification, and enzymatic modification are three major method to achieve this objective. Limited enzymolysis of protein can effectively modify some functional properties, such as the antioxidant activities; moreover, enzymolysis possess advantages of mild reaction condition and high modification efficiency. In addition, as a moderate chemical method, glycosylation has exhibited excellent modification effect of native proteins and thus has been selected as a potential to explore food ingredients. Therefore, this study conducted limited enzymolysis of SPI to obtain different fractions of SPI hydrolysates with different molecular weight and then glycosylation of the obtained SPI hydrolysate fractions with dextran, aiming to investigate the combined modification effects on the general structural properties and antioxidant activities. In fact, reasons for selecting enzymolysis are introduced in L38-42 and L65-66.

3) Why authors used dextran instead of dextrin. Please add some reasons in the manuscript.

Actually, dextran is a commonly used polysaccharide used to conduct the glycosylation to improve the functionalities of native proteins. The reasons for selecting dextran in this study are added in the manuscript now according to this advice (see L67-71).

4) Did you check the glycosylation of soybean protein isolate (without enzymatic hydrolysis)?.

Thank you for this comment. In our previous study, SPI grafted with dextran with a series of molecular weight was already investigated and the corresponding results were published (LWT - Food Science and Technology, 2021, 139: 110588).

5) What is the percentage of glycosylation? Because the changes in the graphs of 1C and 1D do not seem to match.

Degree of grafting (also known as percentage of glycosylation) was not measure to character the reaction degree of glycosylation in this study. On the contrary, the reaction degree is reflected by measuring the content of Amadori products (A304) and and browning degree (A420). As indicated previously, glycosylation is the first stage of the Maillard reaction involving the formation of Amadori products; therefore, A304 can express the reaction degree. In addition, browning degree indicates the proceeding of the final stage of the Maillard reaction. As exhibited in Fig. 1C and 1D, F30 with relatively higher molecular weight possessed higher A304 than F30-10 or F10, indicating the F30 grafting with dextran exhibited greater reaction degree.

6) The explanation of the dot should be included in the description of Figure 2.

Thank you for this meaningful comment. The explanation of the dotted box is added in the caption of Figure 2 now.

7) I don't understand the difference within the dots in Figure 2. Please change to a easier figure to understand..

Thank you for this important advice. The black dotted box is replaced by red dotted box now.

8) Please show the absorption intensity of newly formed C-N bonds by the Maillard reaction (Figure 4).

Thank you for this comment. Based on this comment, the absorption intensity of newly formed C-N bonds was added in Figure 4B-4D in the form of transmittance (%) (L429-432).

9) Authors mentioned endothermic peaks appeared at 60.91 °C and 45.36 °C for F30-10, while for F10, this peak temperature was further reduced to 54.22 °C and 44.62 °C. Those peaks are not found at 0 h in Fig. 5C D. Please explain what these differences are.

In fact, as exhibited in Figure 5C and 5D, a endothermic peak with reduced peak intensity appeared at 43.19 and 46.68 °C compared with the pure SPIH was observed for the F30-10/dextran mixture and F10-dextran mixture, respectively. The difference might be related to the protection of high-molecular-weight dextran in the mixture state. These descriptions are added to the manuscript now.

10) The authors should indicate the units of Ferrous reducing power.

Thank you for this detailed comment. The absorbance at 700 nm of the reactant solution is commonly used to indicate the ferrous reducing power; therefore, the tile of y-axis in Figure 6E and 6F was revised.

11) In this paper, authors produced three different sizes of glycated protein hydrolysates. Are there any ways to utilize the characteristics of each SPHI?

Thank you for your enlightening comment. In fact, in this study, authors’ original intention was to investigate the structural properties and functionalities of modified conjugates formed by combined enzymolysis and glycosylation with dextran. Therefore, the ways to utilize the characteristics of each SPIH include 1) development of functional ingredients enriched hydrolyzed SPI, 2) exploring SPI-based food ingredients with modified functionalities, 3) preparing potential packaging materials for novel food preservation.

Round 2

Reviewer 2 Report

The authors have replied to my comments and suggestions.